# Agreement between Type 2 Diabetes Risk Scales in a Caucasian Population: A Systematic Review and Report

**DOI:** 10.3390/jcm9051546

**Published:** 2020-05-20

**Authors:** Jose Angel Ayensa-Vazquez, Alfonso Leiva, Pedro Tauler, Angel Arturo López-González, Antoni Aguiló, Matías Tomás-Salvá, Miquel Bennasar-Veny

**Affiliations:** 1Department of Nursing, Universidad de Zaragoza, 50009 Zaragoza, Spain; jayensa@unizar.es; 2Primary Care Research Unit, Balearic Islands Health Service, 07002 Palma, Spain; aleiva@ibsalut.caib.es; 3Department of Fundamental Biology and Health Sciences, Balearic Islands University, 07122 Palma, Spain; 4Prevention of Occupational Risk in Health Services, Balearic Islands Health Service, 07003 Palma, Spain; angarturo@gmail.com (A.A.L.-G.); mtomas@dgfun.caib.es (M.T.-S.); 5Research Group on Evidence, Lifestyles and Health Research, Instituto de Investigación Sanitaria Illes Balears, 07122 Palma, Spain; aaguilo@uib.es (A.A.); miquel.bennasar@uib.es (M.B.-V.); 6Department of Nursing and Physiotherapy, Balearic Islands University, 07122 Palma, Spain

**Keywords:** diabetes mellitus, type 2 diabetes, risk scales, risk scores, prediction model, systematic review

## Abstract

Early detection of people with undiagnosed type 2 diabetes (T2D) is an important public health concern. Several predictive equations for T2D have been proposed but most of them have not been externally validated and their performance could be compromised when clinical data is used. Clinical practice guidelines increasingly incorporate T2D risk prediction models as they support clinical decision making. The aims of this study were to systematically review prediction scores for T2D and to analyze the agreement between these risk scores in a large cross-sectional study of white western European workers. A systematic review of the PubMed, CINAHL, and EMBASE databases and a cross-sectional study in 59,042 Spanish workers was performed. Agreement between scores classifying participants as high risk was evaluated using the kappa statistic. The systematic review of 26 predictive models highlights a great heterogeneity in the risk predictors; there is a poor level of reporting, and most of them have not been externally validated. Regarding the agreement between risk scores, the DETECT-2 risk score scale classified 14.1% of subjects as high-risk, FINDRISC score 20.8%, Cambridge score 19.8%, the AUSDRISK score 26.4%, the EGAD study 30.3%, the Hisayama study 30.9%, the ARIC score 6.3%, and the ITD score 3.1%. The lowest agreement was observed between the ITD and the NUDS study derived score (κ = 0.067). Differences in diabetes incidence, prevalence, and weight of risk factors seem to account for the agreement differences between scores. A better agreement between the multi-ethnic derivate score (DETECT-2) and European derivate scores was observed. Risk models should be designed using more easily identifiable and reproducible health data in clinical practice.

## 1. Introduction

Type 2 diabetes (T2D) is a common disease associated with reduced life expectancy and considerable morbidity [1]. Furthermore, T2D is commonly an asymptomatic condition [2] associated with other non-communicable diseases, and causes a high number of hospitalizations and a significant economic impact [3,4]. According to the International Diabetes Federation (Diabetes Atlas 2019), the prevalence of diabetes is increasing worldwide, and it is estimated to increase from 9.3% to 10.9% by 2045, affecting 700 million adults [5]. In addition, about 50% of all people with diabetes worldwide are not diagnosed [1]. Diabetes prevalence varies among regions, ranging from 5% to 20%, with a higher incidence in Oceania, the Caribbean, South Asia, the Middle East, Latin America, and Central Asia [6].

Early detection of people with undiagnosed T2D is an important public health concern, as up to half of people with newly diagnosed T2D present one or more complications when it is diagnosed [7,8]. Several large clinical trials have shown that diabetes could be prevented by recommending lifestyle changes such as advocating physical activity, a healthy low fat diet and weight reduction [3,9,10]. These lifestyle changes have been shown to decrease the risk of diabetes by nearly 60% [9,11,12,13,14]. Thus, policies focusing on diabetes prevention and early identification of T2D in populations at high risk may be worth seriously considering [15]. Furthermore, some studies show how some diabetes risk scores, such as the German Diabetes Risk Score (GDRS) or the Atherosclerosis Risk in Communities model (ARIC 2009), allow the identification of high-risk target groups for cost-effective lifestyle interventions to prevent T2D [16].

The worldwide number of published risk prediction tools for identifying individuals at risk of T2D has greatly increased in the last few years [15,17,18]. However, there is no consensus on which is the best risk score and only a few of them end up being used in clinical practice. Prognostic risk-score models for T2D vary from those including only clinical variables to those with a genetic score and biochemical markers. These risk scores have been derived over a wide range of different populations.

T2D risk scores should be externally validated using data from different settings, populations, and ethnics groups, because generalization outside of the context in which they were designed could affect their performance and therefore, their external usefulness [18]. A review and external validation of commonly used prediction tools including only clinical and conventional biomarkers demonstrated that risk prediction tools work properly in the validation cohorts [19]. However, most of the models overestimate the number of people at high risk. These fitting differences could be explained by differences in population baseline characteristics and in the methodology used [20,21,22,23]. Furthermore, ethnicity or diabetes incidence could play an important role in the differences between risk scores [24].

The aims of this study were to systematically review predictive T2D equations and to analyze the agreement between these prediction risk scores in a cross-section study of Spanish Caucasian workers. Risk scores derived from populations with very different characteristics such as distinct ethnicities and a wide range of diabetes incidence were considered.

## 2. Material and Methods

### 2.1. Search Strategy

Articles that presented new risk prediction models for detecting T2D were identified. A systematic review was performed in Medline, PubMed, CINAHL, and EMBASE databases following PRISMA guidelines [25] looking for articles that reported models or predictive equations for incident diagnosis of T2D until July 2018. The following search string was used: ((“diabetes mellitus” OR “type 2 diabetes” OR “diabetes”) AND (“predictive model” OR “predictive equation” OR “prediction model” OR “prediction rule” OR “risk assessment” OR “risk score”) NOT (“review” OR “bibliography”)). Articles were restricted to English, Portuguese, and Spanish language literature. Reference lists was also verified for relevant citations. The search strategy was performed in cooperation with a research librarian. Unpublished literature was identified through the Information System on Gray Literature in Europe (Open Gray), Conference Proceedings of the Web of Science and ProQuest Dissertations, and Theses Global. 

References of the studies identified by the literature search strategy were imported into EndNote X9 (Clarivate Analytics, Philadelphia, PA, USA) literature management software, and duplicates were removed. One researcher (A.A.) screened for the titles and abstracts of all articles identified by the search string to exclude articles that did not report risk prediction models. After reviewing the retrieved titles, to ensure the quality of the process, two additional authors (J.A.A.-V. and M.B.-V.) independently reviewed the abstracts to select the relevant papers. Each article was randomly assigned to reviewers. Discrepancies between reviewers were solved by a third reviewer (A.L.). To reduce the risk of bias, a pilot exercise was carried out to apply the inclusion criteria in a sample of 10 references.

Study characteristics and study data was managed using Microsoft Excel 2013 (Microsoft Corp, Redmond, WA, USA, www.microsoft.com) and Review Manager software (RevMan version 5.3, Copenhagen, Denmark: The Nordic Cochrane Centre, the Cochrane Collaboration 2014), respectively. A standardized form was used for data extraction, including general information (author(s), journal, location, year, country, and conflict of interest); population characteristics (age, sex, ethnicity, and inclusion and exclusion criteria); and study details (study design, sample size, statistical analysis, and bias). Articles where the model/equation included genetic testing or non-common biomarkers were excluded.

### 2.2. Study Design

A cross-sectional study with Caucasian western European adult workers (aged 20–65 years) was performed. All subjects were from a Spanish Mediterranean area and belonged to different productive sectors (public administration, health department, and the post office). Study design, procedures and reporting followed guidance from the STROBE statement on observational studies [26]. To evaluate the correlation of the retrieved prediction models, the original published prediction models were used (scores or original regression equations). Then, participants’ risk of developing T2D was determined using the following models: diabetes and impaired glucose tolerance score (DETECT-2) [27]; Danish Diabetes Risk Score (DDRS) [28]; data from the epidemiological study on the insulin resistance syndrome score (DESIR) [29]; Cambridge risk score [30]; QDScore [31]; FINDRISC score [32]; EGAT score [33]; Australian T2D risk score (AUSDRISC) [34]; instrument for T2D score (ITD) [35]; atherosclerosis risk in communities score (ARIC) [36]; San Antonio prediction model risk score [37]; Framingham offspring score [38]; diabetes population risk tool score (DPoRT) [39]; scores from Oman [40], India (IDRS) [41], Taiwan [42], and Kyushu island of Japan [43]; and scores from military officers of China [44], and Mauritian Indians [45]. Six predictive models selected in the systematic review were not included to analyze the correlation with the equation because it was impossible to calculate the score in the German Diabetes Risk Score (GDRS) [46] and the modified GDRS [47] because diet was not ascertained in our population. The Tromsø Study [48], the Mauritan Indians risk score [45], the Tehran lipid and glucose studies [49], and the AusDiab [50] general scores were not determined because the original paper did not report enough data to allow proper score calculations. Agreement between results was analyzed as it is indicated in the statistical analysis section.

### 2.3. Participants and Recruitment

Participants in the study were recruited during their periodic health examination in the workplace between January 2008 and December 2010. Every day each worker was assigned a number, and half of the examined workers were randomly selected using a random number table. Thus, from a total population of 130,487 workers, 65,200 were invited to participate in the study. 14,946 (22.9%) refused to participate, leaving the final number of participants standing at 590,424 (77.1%), with 25,510 women (43.2%), and 33,532 men (56.8%). The mean age of the participants in the study was 39.7 years (SD 10.2). All participants were informed of the purpose of this study before they provided written informed consent to participate. Following the current legislation, members of the Health and Safety Committees were informed as well. The study protocol was in accordance with the Declaration of Helsinki and was approved by the Institutional Review Board of the Mallorca Health Management (GESMA). After acceptance, a self-reported complete medical history, including family and personal history, was recorded. The following inclusion criteria were considered: age between 18 and 65 years (working age population), being gainfully employed, and without a previous diagnosis of diabetes.

Subjects who did not meet any of the inclusion criteria and those who refused to participate were excluded from the study.

### 2.4. Samples and Measurements

The methodology used was similar to the one previously reported [51]. Anthropometric measurements were made in the morning at the same time, and according to the recommendations of the International Standards for Anthropometric Assessment (ISAK) [52]. Furthermore, all measurements were performed by well-trained technicians or researchers to minimize coefficients of variation. Body weight (electronic scale Seca 700 scale, Seca GmbH, Hamburg, Germany), height (stadiometer Seca 220 CM Telescopic Height Rod for Column Scales, Seca GmbH, Hamburg, Germany), and abdominal waist circumference using a Lufkin Executive^®^ Thinline, precision 1 mm (Lufkin Executive Thinline, W606PM, Cooper Industries, Lexington, SC, USA) were determined according to the aforementioned recommended techniques. Body mass index (BMI) was calculated as weight (kg) divided by height (m) squared. Waist circumference was measured halfway between the lower costal border and the iliac crest. The measurement was made at the end of a normal expiration while the subject stood upright, with feet together and arms hanging freely at their sides.

Venous blood samples were taken after participants were seated at rest for at least 15 min from the antecubital vein with suitable vacutainers without anticoagulant to obtain serum. Blood samples were taken following a 12 h overnight fast. Concentrations of glucose, cholesterol and triglycerides were measured in serum by standard clinical biochemistry laboratory procedures using an automated hematology analyzer (SYNCHRON CX^®^9 PRO, Beckman Coulter, Brea, CA, USA).

### 2.5. Statistical Analysis

Descriptive analysis was used to report the frequency and distribution of categorical variables, whereas means and standard deviations (SDs) were reported for quantitative variables. The Spearman correlation coefficients were used to analyze the correlation between prediction scores of T2D. Participants were classified as high risk for developing diabetes if the cut-off points were reported in the publication; the following cut-off points were considered for classifying people at high risk: 31 points out of 60 in the Danish Diabetes Risk Score (DDRS); 7 points out of 32 in the Diabetes and Impaired Glucose Tolerance score (DETECT-2); 0.37 out of 1 point in the Cambridge score; a score of over 6 out of 20 in the FINDRISC; over 12 out of 43 in the Australian T2D Risk (AUSDRISK); over 60 out of 100 in the Indian Diabetes Risk Score (IDRS); over 6 out of 17 in the Electric Generating Authority of Thailand Study (EGATS); over 14 out of 49 in the Hisayama study; over 21 out of 40 in the National Urban Diabetes Survey (NUDS); and 55 out of 100 in the instrument for T2D (ITD). Agreement between 2 by 2 charts of participants classified as high risk was assessed using the kappa statistic. Statistical analyses were performed using IBM SPSS Statistics version 24 (SPSS/IBM, Chicago, IL, USA).

## 3. Results

The literature search strategy retrieved 820 original articles, 42 of which met the inclusion criteria considered (Figure 1).

The summarized characteristics of the studies included in the review are shown in Table 1. We selected prediction tools developed from adult or middle adult populations that include predictors generally available in health records (demographics, history of parental diabetes or gestational diabetes, obesity, diet, lifestyle factors, obesity, antihypertensive medication, use of corticoids) as well as conventional biomarkers such as glucose, HDL-cholesterol, LDL-cholesterol, and triglycerides.

There were 26 papers from six continents, 9 prioritizing scores from Europe [28,29,30,31,32,46,47,48,53], 9 from Asia [33,40,41,42,43,44,45,49,54], 2 from Oceania [34,50], 1 from South-America [35], 5 from North-America [36,37,38,39,55] and finally one including populations or participants from 3 continents (Africa, Asia, and Oceania) [27].

Table 2 summarizes the characteristics of the excluded studies: those reporting the calibration of selected prediction tools [56,57,58,59,60,61], performed in an elderly population [62,63], only in men or hypertensive patients [64,65], validation studies [66,67], and those considering scores that include biomarkers or genotype determinations that are not easily available for clinicians [68,69,70,71,72,73].

In the cross-sectional study, risk of developing T2D was determined in a large sample of workers (*n* = 59,042). Correlations between the retrieved prediction models were evaluated using the original published scores or original regression equations. Agreement between predictions risk scores was analyzed as well. Table 3 summarizes the characteristics of participants in the study.

Table 4 shows the Spearman correlation coefficients of the models. There is a wide range of correlation values, from low to high values of correlation. For example, correlations considering the Cambridge score range from 0.560 to 0.898, and correlations including the Framingham offspring range from 0.481 to 0.760. Table 5 shows the agreement (kappa) between risk predicted (high and non-high risk) and Appendix A shows the distribution of the population into high risk and non-high risk groups. Agreements found between the classifications using the different scores ranged from 0.412 (between the ARIC and the DESIR score) to 0.916 (between AUSDRISK and the Hisayama study).

## 4. Discussion

There is great variability between risk prediction models for developing T2D. These prediction models include a wide range of clinical variables and conventional biomarkers, from the most simple models including only age, waist circumference, parental history of diabetes, and physical exercise practice [41], to the most complex including also dietary characteristics [16,46,47], social deprivation measures [31], educational level [48], and ethnicity [31,34,37,39].

The most commonly used risk predictors were age, BMI or obesity, family history of diabetes, and hypertension. There were differences in the weight of the risk predictors included in the equation, the adjusted odds ratio of obesity for undiagnosed T2D varied when comparing different countries from North Europe such as Denmark (≤30 kg/m^2^ vs. <25 kg/m^2^) 4.4 (2.6–7.3) [28] and Finland 2.99 (1.31–6.81) [32] to those from Asia such as Thailand (≤27,5 kg/m^2^ vs. 23 kg/m^2^) 1.74 (1.17–2.60) [33], and China (≤28 kg/m^2^ vs. 24 kg/m^2^) 1.56 (1.03–2.38) [44]. As it can be observed, there were differences in the BMI cut-off point for obesity, which was lower in Asian populations. The prevalence of a risk factor such as obesity and the cut-off points also differed, ranging from 16.3% (≤30 kg/m^2^) in Demark [28] to 6.3% (≤27.5 kg/m^2^) in the derivation cohort from the EGAT study in Thailand [33].

In the present cross-sectional study, the scores of the retrieved prediction equations as well as the Spearman coefficients for the correlations between them were calculated. In the Caucasian population considered, no higher correlations were found between the scores derived from Caucasians and the ones derived from other ethnic groups, prevalence of diabetes, estimated cut-off points or country proximity. However, higher correlations were found between the scores that included only clinical variables than between those that included clinical and conventional biomarkers. Furthermore, higher correlations were found between models that included only hypertension as risk predictors of T2D than between those that included several cardiovascular risk factors (systolic and diastolic blood pressure, cholesterol, triglyceride levels, etc.) There is also poor agreement when models derived from “special populations” such as volunteers [29], Chinese living in Taiwan [42], or military officers [44] were considered.

The percentage of people classified as high risk ranged from 3.1% [35] to 47.1% [54]. These differences could be due to the fact that the cut-off point to classify people at high risk of developing T2D is clearly related to the incidence of diabetes which, in the studies considered, ranged from 1.3% in an adult population in the UK [30] to 26.6% in a Chinese population of military officers in Beijing [44].

Our study suggests that when highly diverse diabetes incidence and ethnicity derived risk predictor models were applied in a Caucasian worker population, poor agreements were achieved. Furthermore, differences in people classified as at high risk were also observed. Agreement did not improve by prevalence of diabetes or country proximity. However, in a validation cohort in a worldwide population (Africa, Asia, Oceania, North America, and Europe) [20], the area under the curve of white Caucasian population behaved similarly, and showed a better prediction between geographically closer countries, showing lower specificity when European developed risk prediction models were applied to African or Asian populations.

Agreement between models did not improve when ethnicity was considered in the models. In this regard, Tanamas [74], in a multi-ethnic cohort validation of highly diverse ethnicity development predictive models, showed a modest influence of the ethnicity in the development cohort in the prediction but there was no evidence that models performed better in populations with a similarity between the development derivation ethnicity and the ethnics in the validation cohort. In this sense, Rosella [24] in a multi-ethnic cohort showed that adding ethnicity did not improve discrimination or the accuracy of predictive models. The causes of the ethnic differences in T2D incidence are not well known. Specifically, the relative contributions of genetic and environmental factors to such differences are largely unknown. Only a few studies in isolated populations have shown evidence on how differences in frequencies of known T2D susceptibility genetic alleles account for ethnic differences [75]. However, research for genetic susceptibility has not been uniform among the world’s ethnic groups. Actually, ethnicity is associated with many other risk factors for T2D that may account for the race/ethnic differences in risk of T2D. These factors include, among others, obesity or overweight, prediabetes condition, diet characteristics, socioeconomic status, area of residence, and environmental contaminants [75,76]. An improved understanding of the impact of these factors on T2D risk should lead to more effective preventive strategies. Performing better designed research must be a goal to understand the ethnic related risk for T2D. Belonging to similar ethnicities or showing similar T2D risk could not be the best way of ascertaining whether a model will perform properly in another population.

Although from the individual risk perspective, ethnicity information could be important, when predicting new cases of diabetes at the population level, detailed ethnic information has not been shown to improve discrimination and accuracy of the model or to identify a significant higher number of diabetics in the population. Therefore, it could be more important to develop models using measurements highly reproducible and available in the clinical practice.

### Limitations

Our systematic review was limited to English, Portuguese, and Spanish language articles; therefore, we may have missed some useful studies. We would like to emphasize that the purpose of the study was to highlight the heterogeneity of the risk of developing diabetes in this population when using different risk prediction models. It was not possible to validate this onset of disease in participants in the present study since they were not followed up.

## 5. Conclusions

Numerous T2D prediction models exist based on readily available health data and provide an adequate but not perfect estimate of the chance of developing T2D in the future. The systematic review of 26 predictive models highlights a great heterogeneity in the risk predictors included and the cut-off points of some risk predictors. A poor reporting of the development procedure of the risk prediction models in terms of describing the data and providing sufficient detail in all steps taking in building the model has been observed. Furthermore, most of the models have not been externally validated. Ethnicity includes intrinsically important genetic and environmental factors related to diabetes onset; however, the evidence is still controversial as regards the influence of ethnicity as an independent risk predictor for T2D onset. Risk prediction models should be derived from the general population and further research is required to improve prediction of T2D.

Differences in diabetes incidence, prevalence, and weight of risk factors seem to account for the agreement differences between scores. In the Caucasian population of workers considered in the present study there is better agreement between the multi-ethnic derivate score (DETECT-2) and European derivate scores. Risk models development should change towards the use of more available and reproductible risk predictors.

## Figures and Tables

**Figure 1 jcm-09-01546-f001:**
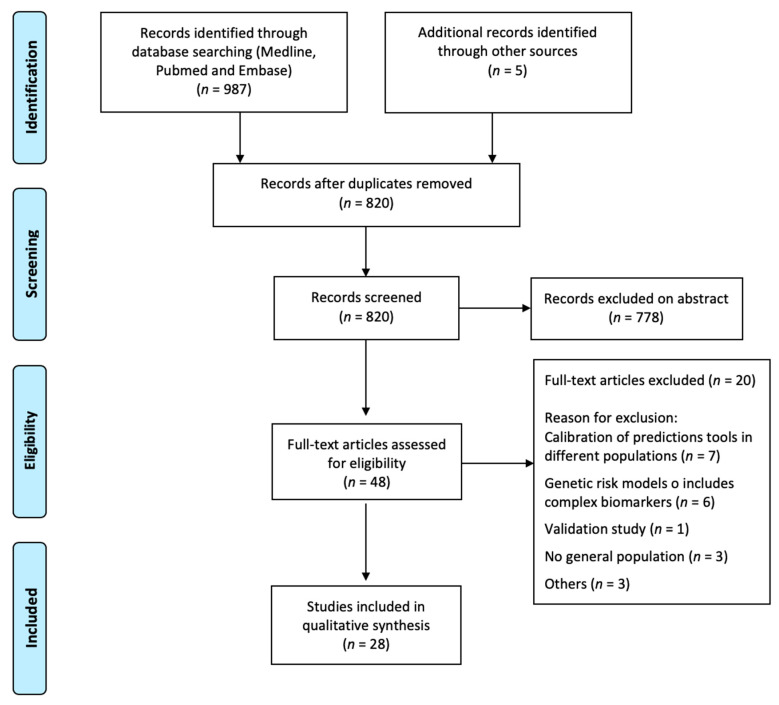
Flow diagram of selected studies.

**Table 1 jcm-09-01546-t001:** Models included for predicting risk of incident type 2 diabetes.

Author/Year	Population	Risk Predictors	Name Risk Diabetes Score	Risk Predictors in the Model	Country	% Incidence Diabetes
Alssema 2011 [27]	Adults	Clinical	DETECT-2	Age, BMI, waist circumference, use of antihypertensive drugs and history of gestational diabetes	Europe, Australia and Africa	4.6
**Europe**						
Glumer 2004 [28]	Adults 30–64 years	Clinical	DDRS	Age, gender, body mass index, known hypertension, regular exercise, and family history of diabetes.	Three European countries (England, Netherlands, Denmark)	4.2
Balkau 2008 [29]	Adults volunteers 30–65 years	Clinical	DESIR	Men: Waist circumference, hypertension, smoking status Women: Waist circumference, family history of diabetes, hypertension	western French	7.5
Schulze 2007 [46]	Adults 35–65 years	Clinical and diet	GDRS	Age, waist circumference, height, history of hypertension, physical inactivity, smoking, consumption of red meat, whole grain bread, coffee, and alcohol	German (Epic-Potsdam)	3.4
Muhlenbrunch 2014 [47]	Adults 35–65 years	Clinical and diet	GDRS-modified	Age, waist circumference, height, history of hypertension, physical, inactivity, smoking, consumption of red meat, whole grain bread, coffee, alcohol and history of diabetes	German (Epic-Potsdam)	2.2
Simmons 2007 [53]	Adults 40–79 years	Clinical and diet	EPIC-Norfolk	Age, gender, physical activity, family history of diabetes, BMI, smoking whole grain bread, fruits	UK (Epic-Norfolk)	1.7
Rahman 2008 [30]	Adults 40–79 years	Clinical	Cambridge	Age, gender, current use of corticosteroids, use of antihypertensive drugs, family history of diabetes, BMI, smoking	UK-Norfolk Cohort	1.3
Hippisley-Cox 2009 [31]	Adults 25–79 years	Clinical and an index of material deprivation	QDScore	Age, gender, ethnicity, BMI, smoking, family history of diabetes, Townsend score, treated hypertension, cardiovascular disease, current use of corticosteroids	England (3–4% of another ethnicity)	3.1
Lindstrom 2013 [32]	Adults 35–64 years	Clinical	FINDRISC	Age, BMI, waist circumference, use of antihypertensive drugs, history of hypertension	Finland National Population Register	4.1
Joseph 2010 [48]	Adults 25–98 years	Clinical and biological	The Tromsø Study	Age, BMI, total cholesterol, triglyceride level, high density lipoprotein cholesterol level, hypertension, family history of diabetes, education, physical inactivity, smoking	North Norway	2.0
**Asia**						
Aekplakorn 2006 [33]	Adult workers 35–55 years	Clinical	EGATS	Age, BMI, waist circumference, hypertension, family history of diabetes in first degree relative	Thailand	11.1
Al-Lawati 2007 [40]	Adults >20 years	Clinical	Omani	Age, BMI, waist circumference, hypertension, family history of diabetes, current hypertension status	Oman	prevalent cases
Jahangiri 2013 [49]	Adults >20 years	Clinical and biological	Teheran study	FPG, 2hPLG, TG, PAS, HDL-C and family history of diabetes	Iran	10.1
Mohan 2005 [41]	Adults >18 years	Clinical	IDRS	Age, waist circumference, hypertension, family history of diabetes, current hypertension status	South Indian	15.5
Ramachandran 2005 [54]	Adults >20 year	Clinical	NUDS	Age, family history of diabetes, BMI, waist circumference, physical activity,	Six cities national survey India	4.6
Gao 2009 [45]	Adults 20–65 years	Clinical	NS	BMI, waist circumference, family history of diabetes	Mauritians Indian	16.5
Liu 2011 [44]	Adults 40–90 years	Clinical and biological	MJLPD	Age, hypertension, history of high blood glucose level, BMI, fasting plasma glucose level, triglyceride level, high density lipoprotein cholesterol level	Military officer in Beijing, China	26.6
Chuang 2011 [42]	Adults >35 years	Clinical	Chinese-DRS	Age, gender, education, alcohol, BMI, waist circumference	Taiwan	6.5
DoI 2012 [43]	Adults 40–79 years	Clinical	Hisayama study	Age, gender, family history of diabetes, abdominal circumference, body mass index, hypertension, regular exercise and current smoking	Japanese Kyushu island	14.7
**Oceania**						
Cameron 2008 [50]	Adults >25 years	Clinical and biological	AusDiab	Age, gender, ethnicity, fasting plasma glucose level, systolic blood pressure, high density lipoprotein cholesterol level, BMI, parental history of diabetes	Australian main Caucasian	3.8
Chen 2010 [34]	Adults >25 years	Clinical	AUSDRISK	Age, gender, BMI, ethnicity, physical inactivity, smoking, history of high blood pressure, use of antihypertensive medication, waist circumference, parental history of diabetes	Australian main Caucasian	5.9
**South-America**						
Guerrero-Romero 2010 [35]	Adults	Clinical and biological	ITD	Age, gender, family history of diabetes, family history of hypertension, family history of obesity, history of gestational diabetes or macrosomia, fasting plasma glucose level, physical inactivity, triglyceride level, systolic or diastolic blood pressure, BMI	México	14.1
**North-America**						
Schmidt 2005 [36]	Adults 45–64 years	Clinical and biological	ARIC	Age, waist circumference, height, systolic blood pressure, family history of diabetes, ethnicity, fasting plasma glucose level, HDL, Triglycerides	US	16.3
Stern 2002 [37]	Adults 25–64 years	Clinical and biologicals	San Antonio	Age, gender, ethnicity, fasting plasma glucose level, systolic blood pressure, high density lipoprotein cholesterol level, BMI, family history of diabetes in first degree relative	US (61% Mexican Americans)	9.2
Wilson 2007 [38]	Middle aged	Clinical and biological	Framingham Offspring	Fasting plasma glucose level, BMI, high density lipoprotein cholesterol level, parental history of diabetes, triglyceride level, blood pressure	US (Framingham)	5.1
Kahn 2009 [55]	Adults	Clinical and biological	ARIC enhanced	Family history of diabetes (mother or father), hypertension, ethnicity, age, alcohol, waist circumference, height, resting pulse, glucose level, triglycerides, HDL cholesterol, uric acid	US	19.0
Rosella 2011 [39]	Adults >20 years	Clinical	DPoRT	Age, ethnicity, BMI, hypertension, immigrant status, smoking, education, cardiovascular disease	Ontario Canada	7.1

DETECT-2: Diabetes and Impaired Glucose Tolerance; DDRS: Danish Diabetes Risk Score; DESIR: Data from the Epidemiological Study on the Insulin Resistance Syndrome; GDRS: German Diabetes Risk Score; EPIC-Norfolk: European Prospective Investigation of Cancer and Nutrition; Cambridge: Cambridge Risk Score; FINDRISC: Finnish Diabetes Risk Score; EGATS: Electric Generating Authority of Thailand Study; IDRS: The Indian Diabetes Risk Score; NUDS: National Urban Diabetes Survey; MJLPD: MJ Longitudinal health-check-up-based Population Database; Chinese-DRS: Chinese Diabetes Risk Score; AusDiab: Australian Diabetes, Obesity and Lifestyle study; AUSDRISK: Australian Type 2 Diabetes RISK; ITD: Instrument for type 2 diabetes (ITD); ARIC: Atherosclerosis Risk in Communities; San Antonio: San Antonio prediction model; Framingham offspring: Framingham Offspring Score; DPoRT: Diabetes Population Risk Tool.

**Table 2 jcm-09-01546-t002:** Models excluded for predicting risk of incident type 2 diabetes.

Author/Year	Name Risk Diabetes	Risk Predictors in the Model	Reason for Exclusion
Alssema 2008 [56]	PREVEND (Modified FINDRISC for Dutch population)	Age, BMI, waist circumference, use of antihypertensive drugs, history of gestational diabetes	Calibration of FINDRISC
Von Eckardstein 2000 [65]	PROCAM score	Age, BMI, hypertension, glucose, family history of diabetes, high density lipoprotein cholesterol level	Only men
Bozorgmanesh 2013 [57]	Modified ARIC-Teheran	Family history of diabetes, systolic blood pressure, waist–height ratio, triglyceride-high density lipoprotein ratio, fasting plasma glucose level, two-hour postprandial plasma glucose level	Calibration of ARIC
Chien 2009 [58]	Cambridge Risk score -Taiwan	Age, BMI, white blood cell count, triglyceride level, high density, lipoprotein cholesterol level, fasting plasma glucose level	Calibration of Cambridge score in Taiwanese population
McNeely 2003 [66]		Age, sex, ethnicity, BMI, systolic blood pressure, fasting plasma glucose level, high density lipoprotein cholesterol level, family history of diabetes in first degree relative	Validation study
Wong 2013 [73]		Sex, age, systolic blood pressure, waist, total cholesterol, HDL-C, triglycerides and HbA1c	Include complex biomarkers
Gupta 2008 [64]		Fasting Plasma Glucose (FPG), history of diabetes and drug or dietary therapy for diabetes. Presence of both impaired FPG (>6 and <7 mmol/L) and glycosuria	Only hypertensive population
Ku 2013 [60]	Findrisk in a Philippine population	Age, BMI, waist circumference, use of antihypertensive drugs, history of hypertension	Calibration Find Risk
Kanaya 2005 [63]		Age, sex, triglyceride level, fasting plasma glucose level	Old adults
Kolberg 2009 [68]	Inter99	Six biomarkers: adiponectin, C reactive protein, ferritin, glucose, interleukin 2 receptor A, insulin	Include complex biomarkers
Chin 2012 [59]	The ARIC predictive model reliably predicted risk of type 2 diabetes in Asian populations	Waist circumference, parental history of diabetes, hypertension, short stature, black race, age, weight, pulse, smoking	Calibration ARIC
Guasch-Ferré 2012 [62]	A risk score to predict type 2 diabetes mellitus in an elderly Spanish Mediterranean population	BMI, smoking status, family history of type 2 diabetes, alcohol consumption and hypertension	Old adults
Vassy 2012 [72]	Genotype prediction of adult type 2 diabetes from adolescence in a multiracial population	Demographics, family history, physical examination, routine biomarkers, and 38 single-nucleotide polymorphism diabetes genotype score	Include a genotype score to predict diabetes
Raynor 2013 [70]	Novel risk factors and the prediction of type 2 diabetes in the Atherosclerosis Risk in Communities (ARIC) study	Adiponectin, leptin, γ-glutamyl transferase, ferritin, intercellular adhesion molecule 1, complement C3, white blood cell count, albumin, activated partial thromboplastin time, factor VIII, magnesium, hip circumference, heart rate, and a genetic risk score	Include genotype
Sun 2009 [61]	The ARIC predictive model reliably predicted risk of type 2 diabetes in Taiwanese population	Waist circumference, parental history of diabetes, hypertension, short stature, black race, age, weight, pulse, smoking	Calibration of ARIC
Meigs 2008 [69]		Age, sex, family history of diabetes, BMI, triglyceride level, fasting plasma glucose level, systolic blood pressure, high density lipoprotein cholesterol level and genotype score	Include genotype
Nichols 2008 [67]		Age, sex, parental history of diabetes, BMI, hypertension or antihypertensive drugs, high density lipoprotein cholesterol level, triglyceride level, fasting plasma glucose level	Validation of the Framingham Offspring Study equations
Urdea 2009 [71]	PreDx diabetes risk score	Levels of adiponectin, C reactive protein, ferritin, glucose, hemoglobinA1c, interleukin 2, insulin	Include complex biomarkers

**Table 3 jcm-09-01546-t003:** Baseline participants’ characteristics (*n* = 59,042).

	Mean (SD)/*n* (%)
Age (years)	39.7 ± 10.2
Men	33,532 (56.8)
Smoking	20,612 (34.9)
Body Mass Index (kg/m^2^)	26.0 ± 4.6
Waist circumference (cm)	82.7 ± 11.6
Systolic blood pressure (mmHg)	120.3 ± 16.0
Diastolic blood pressure (mmHg)	73.4 ± 10.9
Cholesterol (mg/dL)	194.7 ± 37.7
High-density lipoproteins (mg/dL)	52.6 ± 8.5
Low-density lipoproteins (mg/dL)	121.2 ± 37.1
Triglycerides (mg/dL)	107.6 ± 73.0
Fasting plasma glucose (mg/dL)	86.4 ± 12.0

**Table 4 jcm-09-01546-t004:** Correlation between models included for predicting risk of type 2 diabetes.

	DETECT-2	DDRS	DESIR	Cambridge	QDScore	FINDRISC	EGATS	Omani Score	IDRS	NUDS	MJLPD	Chinese DRS	Hisayama Study	AUSDRISK	ITD	ARIC	San Antonio	Framingham Offspring	DPoRT	ARIC Enhanced
DETECT-2	1	0.754 **	0.779 **	0.842 **	0.762 **	0.793 **	0.761 **	0.680 **	0.668 **	0.710 **	0.780 **	0.574 **	0.807 **	0.862 **	0.591 **	0.500 **	0.579 **	0.585 **	0.829 **	0.616 **
DDRS	0.754 **	1	0.515 **	0.881 **	0.926 **	0.872 **	0.832 **	0.881 **	0.866 **	0.911**	0.851**	0.659 **	0.831 **	0.777 **	0.826 **	0.595 **	0.691 **	0.691 **	0.894 **	0.609 **
DESIR	0.779 **	0.515 **	1	0.634 **	0.541 **	0.611 **	0.629 **	0.519 **	0.562 **	0.522 **	0.712 **	0.535 **	0.711 **	0.775 **	0.464 **	0.412 **	0.443 **	0.481 **	0.598 **	0.519 **
Cambridge	0.842 **	0.881 **	0.634 **	1	0.898 **	0.777 **	0.814 **	0.754 **	0.794 **	0.823 **	0.848 **	0.639 **	0.865 **	0.838 **	0.683 **	0.560 **	0.705 **	0.644 **	0.864 **	0.560 **
QDScore	0.762 **	0.926 **	0.541 **	0.898 **	1	0.807 **	0.854 **	0.854 **	0.820 **	0.893 **	0.846 **	0.642 **	0.793 **	0.736 **	0.767 **	0.592 **	0.694 **	0.638 **	0.853 **	0.582 **
FINDRISC	0.793 **	0.872 **	0.611 **	0.777 **	0.807 **	1	0.846 **	0.755 **	0.838 **	0.837 **	0.807 **	0.687 **	0.800 **	0.813 **	0.814 **	0.604 **	0.620 **	0.748 **	0.859 **	0.749 **
EGAT study	0.761 **	0.832 **	0.629 **	0.814 **	0.854 **	0.846 **	1	0.733 **	0.752 **	0.809 **	0.866 **	0.669 **	0.786 **	0.801 **	0.663 **	0.534 **	0.600 **	0.634 **	0.817 **	0.609 **
Omani score	0.680 **	0.881 **	0.519 **	0.754 **	0.854 **	0.755 **	0.733 **	1	0.730 **	0.836 **	0.743 **	0.581 **	0.715 **	0.613 **	0.804 **	0.565 **	0.604 **	0.577 **	0.747 **	0.554 **
IDRS	0.668 **	0.866 **	0.562 **	0.794 **	0.820 **	0.838 **	0.752 **	0.730 **	1	0.851 **	0.806 **	0.631 **	0.830 **	0.761 **	0.784 **	0.580 **	0.651 **	0.650 **	0.787 **	0.645 **
NUDS study	0.710 **	0.911 **	0.522 **	0.823 **	0.893 **	0.837 **	0.809 **	0.836 **	0.851 **	1	0.797 **	0.627 **	0.787 **	0.720 **	0.755 **	0.586 **	0.641 **	0.640 **	0.813 **	0.595 **
MJLPD study	0.780 **	0.851 **	0.712 **	0.848 **	0.846 **	0.807 **	0.866 **	0.743 **	0.806 **	0.797 **	1	0.695 **	0.857 **	0.887 **	0.667 **	0.551 **	0.667 **	0.644 **	0.855 **	0.594 **
Chinese DRS	0.574 **	0.659 **	0.535 **	0.639 **	0.642 **	0.687 **	0.669 **	0.581 **	0.631 **	0.627 **	0.695 **	1	0.704 **	0.680 **	0.698 **	0.610 **	0.679 **	0.788 **	0.681 **	0.650 **
Hisayama study	0.807 **	0.831 **	0.711 **	0.865 **	0.793 **	0.800 **	0.786 **	0.715 **	0.830 **	0.787 **	0.857 **	0.704 **	1	0.916 **	0.686 **	0.537 **	0.663 **	0.675 **	0.811 **	0.602 **
AUSDRISK	0.862 **	0.777 **	0.775 **	0.838 **	0.736 **	0.813 **	0.801 **	0.613 **	0.761 **	0.720 **	0.887 **	0.680 **	0.916 **	1	0.590 **	0.505 **	0.631 **	0.666 **	0.846 **	0.611 **
ITD	0.591 **	0.826 **	0.464 **	0.683 **	0.767 **	0.814 **	0.663 **	0.804 **	0.784 **	0.755 **	0.667 **	0.698 **	0.686 **	0.590 **	1	0.619 **	0.603 **	0.760 **	0.695 **	0.695 **
ARIC	0.500 **	0.595 **	0.412 **	0.560 **	0.592 **	0.604 **	0.534 **	0.565 **	0.580 **	0.586 **	0.551 **	0.610 **	0.537 **	0.505 **	0.619 **	1	0.904 **	0.623 **	0.599 **	0.632 **
San Antonio	0.579 **	0.691 **	0.443 **	0.705 **	0.694 **	0.620 **	0.600 **	0.604 **	0.651 **	0.641 **	0.667 **	0.679 **	0.663 **	0.631 **	0.603 **	0.904 **	1	0.673 **	0.725 **	0.706 **
Framingham offspring	0.585 **	0.691 **	0.481 **	0.644 **	0.638 **	0.748 **	0.634 **	0.577 **	0.650 **	0.640 **	0.644 **	0.788 **	0.675 **	0.666 **	0.760 **	0.623 **	0.673 **	1	0.676 **	0.757 **
DPoRT	0.829 **	0.894 **	0.598 **	0.864 **	0.853 **	0.859 **	0.817 **	0.747 **	0.787 **	0.813 **	0.855 **	0.681 **	0.811 **	0.846 **	0.695 **	0.599 **	0.725 **	0.676 **	1	0.644 **
ARIC enhanced	0.616 **	0.609 **	0.519 **	0.560 **	0.582 **	0.749 **	0.609 **	0.554 **	0.645 **	0.595 **	0.594 **	0.650 **	0.602 **	0.611 **	0.695 **	0.632 **	0.706 **	0.757 **	0.644 **	1

** *p* < 0.001; DETECT-2: Diabetes and Impaired Glucose Tolerance; DDRS: Danish Diabetes Risk Score; DESIR: Data from the Epidemiological Study on the Insulin Resistance Syndrome; Cambridge: Cambridge Risk Score; QDScore: Qresearch Database Score; FINDRISC: Finnish Diabetes Risk Score risk; EGATS: Electric Generating Authority of Thailand Study; IDRS: The Indian Diabetes Risk Score; NUDS: National Urban Diabetes Survey; MJLPD: MJ Longitudinal health-check-up-based Population Database; Chinese-DRS: Chinese Diabetes Risk Score; AUSDRISK: Australian Type 2 Diabetes RISK; ITD: Instrument for type 2 diabetes; ARIC: Atherosclerosis Risk in Communities; San Antonio: San Antonio prediction model; Framingham offspring: Framingham Offspring Score; DPoRT: Diabetes Population Risk Tool.

**Table 5 jcm-09-01546-t005:** Agreement (kappa) between risk predicted (high and non-high risk) by DETECT-2, DDRS, FINDRISC, EGATS, and NUDS scores.

	DETECT-2	DDRS	Cambridge	FINDRISC	EGATS	NUDS	Hisayama	AUSDRISK	ITD	ARIC
DETECT-2	NA	0.501	0.564	0.654	0.497	0.277	0.432	0.531	0.222	0.518
DDRS	0.501	NA	0.638	0.518	0.473	0.293	0.327	0.395	0.271	0.298
Cambridge	0.564	0.638	NA	0.555	0.633	0.414	0.541	0.524	0.187	0.293
FINDRISC	0.654	0.518	0.555	NA	0.664	0.399	0.480	0.645	0.193	0.358
EGATS	0.497	0.473	0.633	0.664	NA	0.562	0.624	0.713	0.121	0.249
NUDS	0.277	0.293	0.414	0.399	0.562	NA	0.540	0.444	0.067	0.123
Hisayama	0.432	0.327	0.541	0.480	0.624	0.540	NA	0.722	0.110	0.240
AUSDRISK	0.531	0.395	0.524	0.645	0.713	0.444	0.722	NA	0.133	0.316
ITD	0.222	0.271	0.187	0.193	0.121	0.067	0.110	0.133	NA	0.281
ARIC	0.518	0.298	0.293	0.358	0.249	0.123	0.240	0.316	0.281	NA

DETECT-2: Diabetes and Impaired Glucose Tolerance; DDRS: Danish diabetes Risk score; Cambridge: Cambridge Risk Score; FIDRISC: Finnish Diabetes Risk Score risk; EGATS: Electric Generating Authority of Thailand Study; NUDS, National Urban Diabetes Survey; Hisayama: Hisayama study; AUSDRISK: Australian Type 2 Diabetes RISK; ITD: Instrument for type 2 diabetes; ARIC: Atherosclerosis Risk in Communities. NA: Non-Applicable.

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
