# Peer review of "Agreement between Type 2 Diabetes Risk Scales in a Caucasian Population: A Systematic Review and Report"

_jcm, 2020, doi:10.3390/jcm9051546_

Round 1
Reviewer 1 Report
In this manuscript the authors have conducted a systematic review of all the published prediction risk models of type 2 diabetes. In addition they utilized a large cohorts of Spanish workers in order to evaluate if such risk scores could be consistently applied to such population.
The aim is sound, the systematic review is well conducted and the literature search rigorous, but it is not really clear what have they done with their large population, please better explain it in methods.
In addition, the presentation of the correlation results is very confusing. It is not easy for a reader to understand only by looking at the correlation tables. Especially for table 5, the authors have to find a better way to show the distribution of at risk individuals in their populations according to the different risk score.
Although the study design (cross-sectional) do not allow to speculate too much on the better risk score in their population, which risk score would they have chosen?
The authors, should add a better explanation of the differences among risk scores derived from non-Caucasian ethnicities.
Author Response
Response to Reviewer 1 Comments
Point 1: In this manuscript the authors have conducted a systematic review of all the published prediction risk models of type 2 diabetes. In addition they utilized a large cohorts of Spanish workers in order to evaluate if such risk scores could be consistently applied to such population.
The aim is sound, the systematic review is well conducted and the literature search rigorous, but it is not really clear what have they done with their large population, please better explain it in methods.
Response 1:We are very grateful to the reviewer for their encouragement. Thank you ever so much for your comments. It is now included in the “study design” section (page 3, lines 124-140), together with information in other sections, a more clear explanation of the work done with the large population. The second aim of the present study, or the second part of the aim, was to determine the agreement between the risk predictors found in the review of the literature when these predictors were applied to a Caucasian population. Therefore, this agreement was analyzed in the large sample of participants recruited. After measuring or determined all parameters required, we determined most of the risk predictors found in the review of the literature in the large sample. Using values obtained the agreement analysis was performed as it is indicated in the “statistical analysis” section (page 4, lines 175-189).
Point 2: In addition, the presentation of the correlation results is very confusing. It is not easy for a reader to understand only by looking at the correlation tables. Especially for table 5, the authors have to find a better way to show the distribution of at risk individuals in their populations according to the different risk score.
Response 2: Many thanks for the suggestion. We agree with the reviewer, we have modified the Table 5 and now only kappa values from the agreement of the distribution of participants at high and non-high risk between scores is shown in the new table 5 (page 14). A new table, Table S1 as supplementary material, with the information of the old Table 5 has been also included in the revised version of the manuscript (pages 15-16).
Point 3: Although the study design (cross-sectional) do not allow to speculate too much on the better risk score in their population, which risk score would they have chosen?
Response 3: Thank you for this question. Our study found a high heterogeneity in the prediction of the risk to develop type 2 Diabetes, but not allow to conclude which risk score predict better.
Our findings show that probably there is overestimation of people at risk or high risk for T2D. Occurrence factors for T2D could be different in high incidence populations compared to low incidence populations, and genetic factors could be more important in low incidence populations. Furthermore, BMI cut-off points are different depending on the ethnic group (for example, Asian risk equations use lower cut-off points). To answer the question, all these factors must be taken into account. In addition, we have observed that most of the predictive equations have not been externally validated and many of the models do not use easily identifiable and reproducible health data in clinical practice, which limits their usefulness.
There is no an universal ideal risk score, as the utility of any score depends not merely on its statistical properties but also on its context of use. In clinical or public health practice, one would perhaps prefer to use a model including only a limited number of predictors based on non-invasive tests with the highest performance.
In our personal opinion, the risk scores we would choose to predict the onset of the diabetes will be the FINDRISK or the QDSscore. They both are well-known risk scores, with good predictive properties and they have been validated in different studies.
Point 4: The authors, should add a better explanation of the differences among risk scores derived from non-Caucasian ethnicities.
Response 4: Thanks for your comment. Following your suggestion, we have added more discussion about differences among risk scores ethnicities. (page 16 lines 310-313 and page 17 lines 314-327): “The causes of the ethnic differences in T2D incidence are not well known. Specifically, the relative contributions of genetic and environmental factors to such differences are largely unknown. Only a few studies in isolated populations has shown evidence about differences in frequencies of known T2D susceptibility genetic alleles account for ethnic differences (Golden, 2019). However, research for genetic susceptibility has not been uniform among the world’s ethnic groups. Actually, ethnicity is associated with many other risk factors for T2D that may account for the race/ethnic differences in risk of T2D. These factors include, among others, obesity or overweight, prediabetes condition, diet characteristics, socioeconomic status, area of residence and environmental contaminant (Thakarakkattil, 2020)”
Golden, S. H.; Yajnik, C.; Phatak, S.; Hanson, R. L.; Knowler, W. C., Racial/ethnic differences in the burden of type 2 diabetes over the life course: a focus on the USA and India. Diabetologia 2019, 62, (10), 1751-1760.
Thakarakkattil Narayanan Nair, A.; Donnelly, L. A.; Dawed, A. Y.; Gan, S.; Anjana, R. M.; Viswanathan, M.; Palmer, C. N. A.; Pearson, E. R., The impact of phenotype, ethnicity and genotype on progression of type 2 diabetes mellitus. Endocrinol Diabetes Metab 2020, 3, (2), e00108

Reviewer 2 Report
This systematic review and a validation study of several diabetes risk scores is well done and the validation is based on a cross-sectional analysis in a large Spanish population sample. The paper is well written. The findings highlight the heterogeneity of the prediction capability among different risk scores.
Various scores have been based on different age ranges, and they may have been different than that in the Spanish study population. It would have been useful to carry out a sensitivity analysis using the same age range than in the original populations.
It would be interesting to know possible reasons for poor performance between certain risk scores and also some risk scores when tested in the Spanish study population.
Reference numbering in the text does not match with the numbers in the reference list, and some references are missing in the list. Also, it would be important to include reference numbers in Tables 1 and 2.
Minor comments:
Do not use the word “patient” when discussing about people in the study populations.
Some references have errors in titles or journal names.
Author Response
Response to Reviewer 2 Comments
Point 1: This systematic review and a validation study of several diabetes risk scores is well done and the validation is based on a cross-sectional analysis in a large Spanish population sample. The paper is well written. The findings highlight the heterogeneity of the prediction capability among different risk scores.
Various scores have been based on different age ranges, and they may have been different than that in the Spanish study population. It would have been useful to carry out a sensitivity analysis using the same age range than in the original populations.
Response 1: Thank you ever so much for your comments. We agree with the reviewer that the populations from the scores derivate and were validated very heterogeneous and the distribution of age is very different from scores with quite limited age range as the ARIC (45 to 64 years) to a wider age range as the DETECT-2. However, we found that the analysis of the correlation of two scores limited to the age range of the populations from which the score derivate includes only the individuals that meet both age criteria. For instance, if we compare the ARIC (45 to 64 years) and the EPIC-Norfolk (40 to 79 years) we only compare individual in the range 45 to 64 years because they are the only ones meeting both criteria. We believe that, using this approach, an arbitrary criterion for the risk scores not limited by the age range but for the age range of the pair of risk scores compared is generated.
Point 2: It would be interesting to know possible reasons for poor performance between certain risk scores and also some risk scores when tested in the Spanish study population.
Response 2: The cross-sectional nature of the study does not allow to conclude about performance of the risk scores. We observed that predicted risk does not fit properly when this risk is determined using different equations or models. Probably, one of the main reasons for this is that equations and risk models have been performed from a specific population rather than using different populations and/or people covering a wide range of values for each parameter or characteristic included in the model.
Point 3: Reference numbering in the text does not match with the numbers in the reference list, and some references are missing in the list.
Also, it would be important to include reference numbers in Tables 1 and 2.
Response 3: Sorry for the mistake, we have amended it. All reference numbering in the text have been reviewed and missing references have been added.
Many thanks for the suggestion.We have added the reference numbers in Tables 1 and 2.
Point 4: Minor comments: Do not use the word “patient” when discussing about people in the study populations.
Response 4: Thank you very much, we appreciate this recommendation. We have changed the term “patient” to “people, population or participants” when we discuss about people in the study populations (page 2, lines 79-80; page 4, lines 179 and 181; page 16, lines 291, 292 and 298).
Point 5: Some references have errors in titles or journal names.
Response 5: We agree with the reviewer and the entire of references section has been reviewed and mistakes amended

Round 2
Reviewer 2 Report
The authors have addressed the reviewers' comments adequately.